# CRISPR/Cas9-mediated glycolate oxidase disruption is an efficacious and safe treatment for primary hyperoxaluria type I

Nerea Zabaleta[1], Miren Barberia[1], Cristina Martin-Higueras [2], Natalia Zapata-Linares[3], Isabel Betancor[2], Saray Rodriguez[3], Rebeca Martinez-Turrillas[3], Laura Torella[1], Africa Vales[1], Cristina Olagüe[1], Amaia Vilas-Zornoza[4], Laura Castro-Labrador[4], David Lara-Astiaso[4], Felipe Prosper[3,4,5], Eduardo Salido [2], Gloria Gonzalez-Aseguinolaza[1] & Juan R. Rodriguez-Madoz [3]

CRISPR/Cas9 technology offers novel approaches for the development of new therapies for many unmet clinical needs, including a significant number of inherited monogenic diseases. However, in vivo correction of disease-causing genes is still inefficient, especially for those diseases without selective advantage for corrected cells. We reasoned that substrate reduction therapies (SRT) targeting non-essential enzymes could provide an attractive alternative. Here we evaluate the therapeutic efficacy of an in vivo CRISPR/Cas9-mediated SRT to treat primary hyperoxaluria type I (PH1), a rare inborn dysfunction in glyoxylate metabolism that results in excessive hepatic oxalate production causing end-stage renal disease. A single systemic administration of an AAV8-CRISPR/Cas9 vector targeting glycolate oxidase, prevents oxalate overproduction and kidney damage, with no signs of toxicity in $Agxt1^{-/-}$ mice. Our results reveal that CRISPR/Cas9-mediated SRT represents a promising therapeutic option for PH1 that can be potentially applied to other metabolic diseases caused by the accumulation of toxic metabolites.

[1] Gene Therapy and Regulation of Gene Expression Program, Center for Applied Medical Research (CIMA), University of Navarra, IdiSNA, Pamplona 31008, Spain. [2] Hospital Universitario de Canarias, Universidad La Laguna, CIBERER, Tenerife 38320, Spain. [3] Regenerative Medicine Program, Center for Applied Medical Research (CIMA), University of Navarra, IdiSNA, Pamplona 31008, Spain. [4] Advance Genomics Laboratory, Oncohematology Program, Center for Applied Medical Research (CIMA), University of Navarra, IdiSNA, Pamplona 31008, Spain. [5] Area of Cell Therapy, Clínica Universidad de Navarra, University of Navarra, IdiSNA, Pamplona 31008, Spain. These authors jointly supervised this work: Gloria Gonzalez-Aseguinolaza, Juan R. Rodriguez-Madoz. Correspondence and requests for materials should be addressed to E.S. (email: esalido@ull.es) or to G.G.-A. (email: ggasegui@unav.es) or to J.R.R.-M. (email: jrrodriguez@unav.es)

Primary hyperoxalurias (PH) are a group of autosomal recessive metabolic disorders characterized by defects in enzymes involved in glyoxylate metabolism. The disease is caused by oxalate overproduction, an end-product of glyoxylate metabolism with no biological role in mammals that is produced in the liver and excreted by the kidney[1,2]. Under normal circumstances oxalate is produced at low levels and continuously removed by the kidneys from blood and excreted in urine. However, in PH patients renal accumulation of oxalate results in urolithiasis, nephrocalcinosis and progression to end-stage renal disease (ESRD)[1,2]. There are three PH forms (PH1, 2, and 3), with PH1 being the most common (around 70–80% of all PH patients)[3,4] and severe subtype. PH1 is a life-threatening disease caused by mutations in the *AGXT* gene leading to impaired activity of the hepatic enzyme alanine-glyoxylate aminotransferase (AGT)[5], which catalyzes glyoxylate conversion to glycine (Fig. 1a). AGT malfunction results in progressive decrease of glomerular filtration rate (GFR), and ultimately leads to ESRD and, if untreated, death in most patients[6].

Early diagnosis and initiation of conservative therapy are critical in preserving adequate renal function for as long as possible[7]. Existing treatments aim to ameliorate disease manifestations, but once the renal function is compromised, combined kidney and liver transplantation, with a better survival rate than isolated kidney transplant[8,9], is the preferred option[10,11]. Thus, new therapeutic options are urgently needed. Novel molecular approaches are currently under investigation, including the use of oxalate degrading enzymes, correction of AGT mistargeting (applicable only to a specific population of patients), as well as gene and cell therapies[12–17]. An attractive therapeutic strategy for diseases associated with the accumulation of toxic metabolic products, like PH1, is based on a substrate reduction therapy (SRT)[18–20]. In the context of PH1, inhibition of glycolate oxidase (GO), the enzyme involved in the production of glyoxylate from glycolate, has been postulated as a promising strategy[21–24] (Fig. 1a). In fact, GO deficiency is a very rare disorder, with only two published cases[25,26], showing asymptomatic glycolic aciduria as the only consequence of lack of GO activity. Moreover, in mice,

GO deficiency (*Hao1*$^{-/-}$) results in an increased production of glycolate, which is a highly soluble molecule that can be eliminated in urine without inducing kidney damage or other related toxicity[21]. Furthermore, with respect to the hyperoxaluric *Agxt1*$^{-/-}$ mice, mice genetically deficient in both *Agxt1* and *Hao1* genes (*Agxt1*$^{-/-}$/*Hao1*$^{-/-}$) have normal levels of oxalate excretion in the absence of pathological findings[21]. In addition, pharmacological GO inhibition using small molecules, or GO silencing by small-interfering RNAs (siRNA), can also achieve a significant reduction in oxalate excretion in *Agxt1*$^{-/-}$ mice[22–24]. The safety and efficacy of GO silencing in PH1 patients is currently being investigated in ongoing clinical trials (Alnylam Pharmaceuticals 2016, NCT02706886). However, although the use of small molecules or siRNAs can be effective to block disease-related pathways, they have limitations such as the requirement of multiple administrations for long-term effect[24], incomplete inhibition of the target enzyme, compliance and potential interactions of the small molecules with other drugs/treatments[27].

The latest advances in genome editing using sequence-specific nucleases, particularly the bacterial type II clustered regularly interspaced short palindromic repeats/Cas9 (CRISPR/Cas9) systems[28–30], provide essential tools that enable substantial progress in the development of new therapies[31–34]. The CRISPR/Cas9 system consists of a programmable single-guide RNA (sgRNA), which directs Cas9 endonuclease to the desired genomic locus followed by a consensus sequence (protospacer adjacent motif, or PAM) to induce double-strand breaks (DSBs) into the DNA. DSBs can be repaired by homologous recombination (HDR) during the S/G2 phase of the cell cycle or by the error-prone non-homologous end joining (NHEJ) machinery, which can be exploited to induce insertions or deletions (indels) for gene inactivation. Indeed, therapeutic approaches based on in vivo CRISPR/Cas9 genome editing using both HDR and NHEJ have been recently developed for different diseases, such are hemophilia[35], Duchenne muscular dystrophy[36–38], retinitis pigmentosa[39] or congenital metabolic diseases like hereditary tyrosinemia type I[40,41], urea cycle disorders[42] or

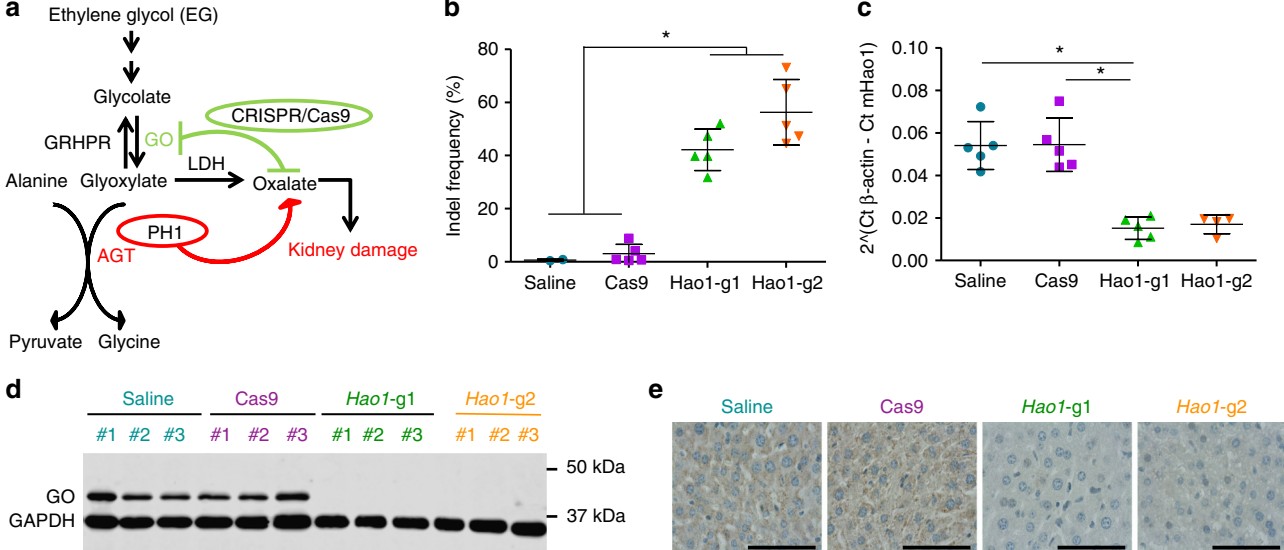

**Fig. 1** Efficient GO inhibition using CRISPR/Cas9 in PH1 animals. **a** Schematic representation of the CRISPR/Cas9-mediated SRT strategy targeting the *Hao1* locus. **b** Editing efficiency measured by TIDE in 12–14-week-old PH1 animals 4 weeks after treatment with saline (*n* = 5), Cas9 (*n* = 5), *Hao1*-g1 (*n* = 5) and *Hao1*-g2 (*n* = 5). **c** Quantification of *Hao1* mRNA expression levels by RT-qPCR in animals treated as in (**b**). Data are presented as mean ± SEM and Kruskal-Wallis statistical test was used to evaluate differences between groups. **d** Western blot analysis of GO protein levels in representative PH1 animals treated with saline, Cas9, *Hao1*-g1, and *Hao1*-g2. GAPDH was used as loading control. **e** Representative IHC images of liver sections stained for GO, from PH1 animals treated with saline, Cas9, *Hao1*-g1, and *Hao1*-g2. Scale bar: 100 μm. *$p < 0.05$

hypercholesterolemia[43]. Although correcting the disease-causing mutations by HDR represents the ideal approach for the treatment of inherited liver metabolic diseases like PH1, this strategy is still very inefficient, due to the low mitotic index of hepatocytes and the bias for the NHEJ pathway to repair DSBs. Furthermore, since in PH1 the hepatocytes remain healthy and there is no liver damage, the corrected hepatocytes have no advantage and will not be selected to repopulate the liver as has been demonstrated for other diseases like tyrosinemia[40]. We postulate that in vivo CRISPR/Cas9-mediated SRT to genetically inactivate non-essential enzymes of the glyoxylate pathway, such as GO, could be an efficient therapeutic approach for treating PH1 (Fig. 1a).

In this work, we established an AAV-based CRISPR/Cas9 system for targeted gene disruption of murine *Hao1* that was evaluated in a preclinical mouse model of PH1[16]. Our studies showed that a single-dose administration of the therapeutic vector resulted in long-term inhibition of hepatic GO, which lead to the reduction of urine oxalate excretion to normal levels and prevented nephrocalcinosis formation with an absence of toxic effects.

## Results

**Efficient in vivo GO inhibition using CRISPR/Cas9**. To develop a CRISPR/Cas9-mediated in vivo GO inhibition strategy as potential SRT for PH1, two single guide RNAs (sgRNA) targeting exonic regions of the murine *Hao1* gene were designed and selected based on location and predicted high on-target and low off-target efficiency (Supplementary Table 1). A single AAV8 vector, was used to deliver *Staphylococcus aureus* Cas9 (SaCas9) and the *Hao1* specific sgRNA to the liver of *Agxt1*$^{-/-}$ mice. To restrict genome editing to hepatocytes, the expression of SaCas9 was controlled by the liver-specific thyroxine-binding globulin (TBG) promoter[44,45] (Supplementary Fig. 1a). Gene editing efficiency of the AAV8-SaCas9-*Hao1*-sgRNA system was evaluated in 12–14-week-old PH1 male mice, which were treated intravenously with AAV8-SaCas9-*Hao1*-sgRNA1 (*Hao1*-g1), AAV8-Cas9-*Hao1*-sgRNA2 (*Hao1*-g2) or AAV8-SaCas9 without sgRNA (Cas9), at a dose of $5 \times 10^{12}$ vg/kg. A group of animals that received saline were used as controls. Mice were sacrificed 4 weeks after treatment and vector genome copies and SaCas9 expression levels were evaluated, with no differences among the groups that received the vectors (Supplementary Fig. 1b and 1c). Cleavage efficacy was analyzed by SURVEYOR assay and indel frequencies were measured by TIDE[46] (Fig. 1b and Supplementary Fig. 1d). An average of $42.14 \pm 7.81\%$ and $56.32 \pm 12.33\%$ of indels were detected in the livers of the animals having received *Hao1*-g1 and *Hao1*-g2, respectively. Moreover, *Hao1* transcription levels were reduced in the animals receiving the therapeutic vectors (Fig. 1c). In accordance, Western blot analysis showed a dramatic reduction of GO protein expression (Fig. 1d). Furthermore, immunohistochemical analysis revealed small numbers of isolated GO-positive cells in AAV8-SaCas9-*Hao1*-sgRNA-treated animals, while control animals had a homogenous staining of the hepatocytes (Fig. 1e).

**Characterization of CRISPR/Cas9-mediated genome editing**. In order to better characterize genome editing efficiency and variant distribution in the liver of animals treated with the therapeutic vectors, the *Hao1* targeted locus was amplified by PCR to generate barcoded libraries that were analyzed by next-generation sequencing (NGS). Deep sequencing of animals having received *Hao1*-g1 or *Hao1*-g2 vectors ($n = 5$ per sgRNA) confirmed previous results with an indel average of $48.46 \pm 3.41\%$ for *Hao1*-g1 and $52.81 \pm 3.85\%$ for *Hao1*-g2 (Fig. 2a and Table 1). Interestingly, cleavage efficiency increased significantly when

analyzed in purified hepatocytes, as expected due to the hepato-specific expression of SaCas9 (Supplementary Fig. 1e). The most common mutations for both sgRNAs were small indels under 10 bp (average of 80.66% for *Hao1*-g1 and 85.03% for *Hao1*-g2), occurring around positions $-3$ to $+3$ relative to the cleavage site (Fig. 2b and Supplementary Fig. 2). Interestingly, indel distribution was different between sgRNAs, with *Hao1*-g1 being prone to introduce insertions of 1 bp, while *Hao1*-g2 more frequently induced 2 bp deletions (Fig. 2b, c). Moreover, although insertions were predominantly introduced at position $+1$ for both sgRNAs, deletions were more widely distributed (Supplementary Fig. 2). These differences in indel distribution might be due to the different target sites selected for each sgRNA, since *Hao1*-g1 targets the 5'-end of the *Hao1* exon 2 and *Hao1*-g2 is directed to the middle of the same exon. More importantly, despite the differences in indel distribution, both sgRNAs introduced frameshift mutations (88.45% and 80.42%, respectively), which explained the reduction in protein levels (Fig. 2b).

**Therapeutic efficacy of CRISPR/Cas9-mediated SRT**. We further evaluated the efficacy of CRISPR/Cas9-mediated GO inhibition as a therapeutic treatment for PH1, aiming to reduce urine oxalate levels and to prevent kidney damage (Fig. 3a). Thus, PH1 male mice were treated as described above with therapeutic and control vectors. Since *Agxt1*$^{-/-}$ mice become hyperoxaluric with respect to control WT littermates from 3 months of age onwards[16], we analyzed oxaluria levels in 24 h urine samples collected 4 months after AAV administration (6–7 month-old animals). Mice treated with *Hao1*-g1 and *Hao1*-g2 vectors presented with significantly lower levels of urine oxalate than PH1 control groups (Cas9 and saline) and levels similar to those of WT animals. This indicated that CRISPR/Cas9-mediated GO inhibition reduced oxalate accumulation (Fig. 3b). As expected, urine glycolate levels were increased as a result of GO reduction (Fig. 3b). Since oxaluria is greatly increased after overload of the glyoxylate pathway, therapeutic efficacy was also analyzed in PH1 animals challenged during 7 consecutive days with ethylene glycol (EG), a precursor of glyoxylate that increases oxalate production[17,22]. Urine oxalate levels, measured on days 3 and 7 of EG challenge, were significantly lower in the animals that received *Hao1*-g1 and *Hao1*-g2 than the PH1 control groups (Cas9 and Saline), and more importantly, similar to levels observed in WT animals (Fig. 3c). Again, as expected, significantly higher urine glycolate levels were observed in treated animals (Fig. 3d). No difference in water intake was observed between the groups, indicating a homogenous EG challenge (Supplementary Fig. 3a). Previous studies have shown that EG challenge in WT mice, which are capable of eliminating high concentrations of oxalate in the urine, is well tolerated without any apparent toxic effect and weight loss[16]. Interestingly, in contrast to Cas9 and saline controls, no body-weight loss was observed during the challenge in animals treated with the therapeutic vectors, indicating the efficiency of the CRISPR/Cas9-mediated SRT treatment (Fig. 3e).

**CRISPR/Cas9-mediated SRT prevented nephrocalcinosis**. Since *Agxt1*$^{-/-}$ mice develop varying degrees of kidney calcium oxalate deposits when challenged with EG[16], the capacity of the treatment to prevent nephrocalcinosis was also analyzed. Thus, PH1 animals were treated with the therapeutic vectors as described above, and two groups of control animals (saline and Cas9) were included. To induce nephrocalcinosis animals were challenged with EG for a longer period of time (10 days), and kidney histology was analyzed. While control animals showed varying degrees of calcium oxalate (CaOx) deposits in their kidneys, with 3 out of 8 animals developing severe or moderate

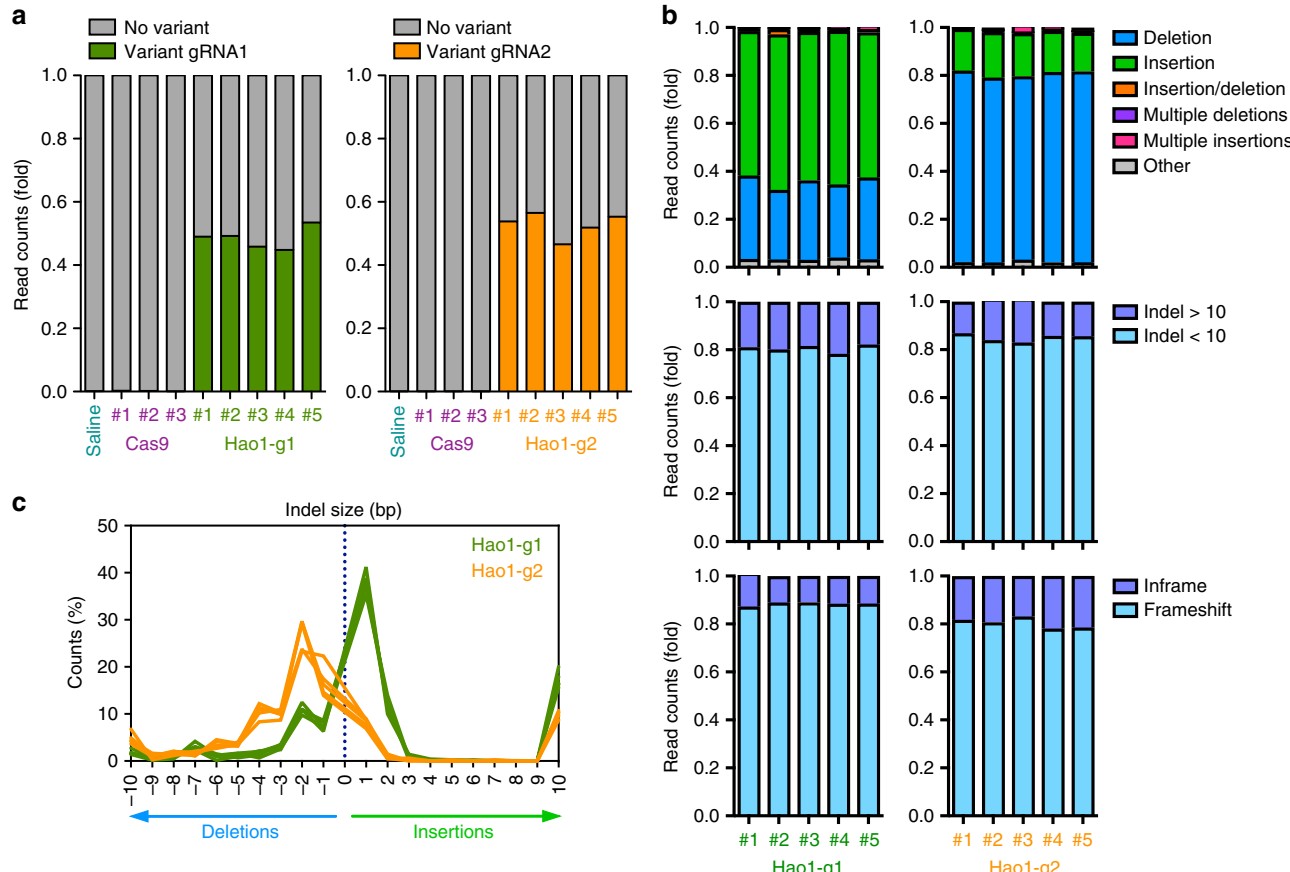

**Fig. 2** Characterization of CRISPR/Cas9-mediated *Hao1* gene editing. Deep sequencing was performed 1 month after treatment on the DNA from livers of 12–14-week-old PH1 animals treated with *Hao1*-g1 ($n = 5$) and *Hao1*-g2 ($n = 5$), as well as Cas9 ($n = 3$) and saline ($n = 1$). **a** Frequency of CRISPR/Cas9 introduced variants in the *Hao1* gene of individual animals. **b** Characterization of the variants according to their type, size, and frameshift potential of each animal treated with *Hao1*-g1 and *Hao1*-g2. **a, b** Each bar represents an individual mouse. **c** Frequency distribution of indel size in base pairs (bp) for each animal treated with *Hao1*-g1 (green lines) and *Hao1*-g2 (orange lines). Each line represents an individual mouse

| Table 1 Average percentage of variants in on-target and off-target sites | | | | | | | | | | |
|---|---|---|---|---|---|---|---|---|---|---|
| **Target Site** | **Time after treatment** | **Group** | **ONT** | **OFT1** | **OFT2** | **OFT3** | **OFT4** | **OFT5** | **OFT6** | **OFT7** |
| Hao1-g1 | 4 weeks | Saline ($n = 1$) | 0.14 | 0.12 | 0.1 | 0.03 | 0.08 | 0.01 | 0.26 | 0.11 |
| | | Cas9 ($n = 3$) | 0.18 | 0.17 | 0.09 | 0.06 | 0.08 | 0.04 | 0.3 | 0.08 |
| | | Hao1-g1 ($n = 5$) | 48.46 | 0.18 | 0.08 | 0.05 | 0.08 | 0.04 | 0.29 | 0.08 |
| | 6 months | Saline ($n = 1$) | 0.14 | 0.12 | 0.1 | 0.03 | 0.08 | 0.01 | 0.26 | 0.11 |
| | | Cas9 ($n = 3$) | 0.24 | 0.18 | 0.07 | 0.06 | 0.07 | 0.04 | 0.28 | 0.07 |
| | | Hao1-g1 ($n = 5$) | 60.57 | 0.19 | 0.08 | 0.05 | 0.07 | 0.03 | 0.32 | 0.1 |
| Hao1-g2 | 4 weeks | Saline ($n = 1$) | 0.14 | 0.12 | 0.1 | 0.03 | 0.08 | 0.01 | 0.26 | 0.11 |
| | | Cas9 ($n = 3$) | 0.08 | 0.07 | 0.09 | 0.09 | 0.06 | 0.07 | 0.1 | 0.11 |
| | | Hao1-g2 ($n = 5$) | 52.81 | 0.08 | 0.09 | 0.08 | 0.06 | 0.06 | 0.13 | 0.11 |
| | 6 months | Saline ($n = 1$) | 0.14 | 0.12 | 0.1 | 0.03 | 0.08 | 0.01 | 0.26 | 0.11 |
| | | Cas9 ($n = 3$) | 0.08 | 0.07 | 0.09 | 0.1 | 0.08 | 0.06 | 0.11 | 0.12 |
| | | Hao1-g2 ($n = 5$) | 60.13 | 0.07 | 0.08 | 0.08 | 0.07 | 0.07 | 0.1 | 0.12 |

*ONT* on-target, *OFT* off-target

nephrocalcinosis (3.2–6.2% of kidney area presenting CaOx crystals, Fig. 3f and Supplementary Fig. 3b), only 1 out of 10 animals treated with the therapeutic vectors displayed mild CaOx deposition (0.7% of kidney area presenting CaOx crystals, Fig. 3f and Supplementary Fig. 3b). These results indicate that our CRISPR/Cas9-mediated SRT strategy was able not only to reduce the urine oxalate levels, but also to protect against kidney stone formation caused by prolonged hyperoxaluria in a PH1 mouse model.

**Safety of CRISPR/Cas9-mediated SRT therapy.** One of the main concerns regarding therapies involving CRISPR/Cas9 is the potential genotoxic effect due to non-specific cleavage and off-target indel generation. To address this question, we performed targeted deep sequencing at the on-target site as well as the predicted off-target regions (top 7 score, Benchling) for each sgRNA in the livers of treated animals 6 months after AAV administration (Supplementary Table 2). In comparison to the data obtained 4 weeks after vector injection, we observed slightly

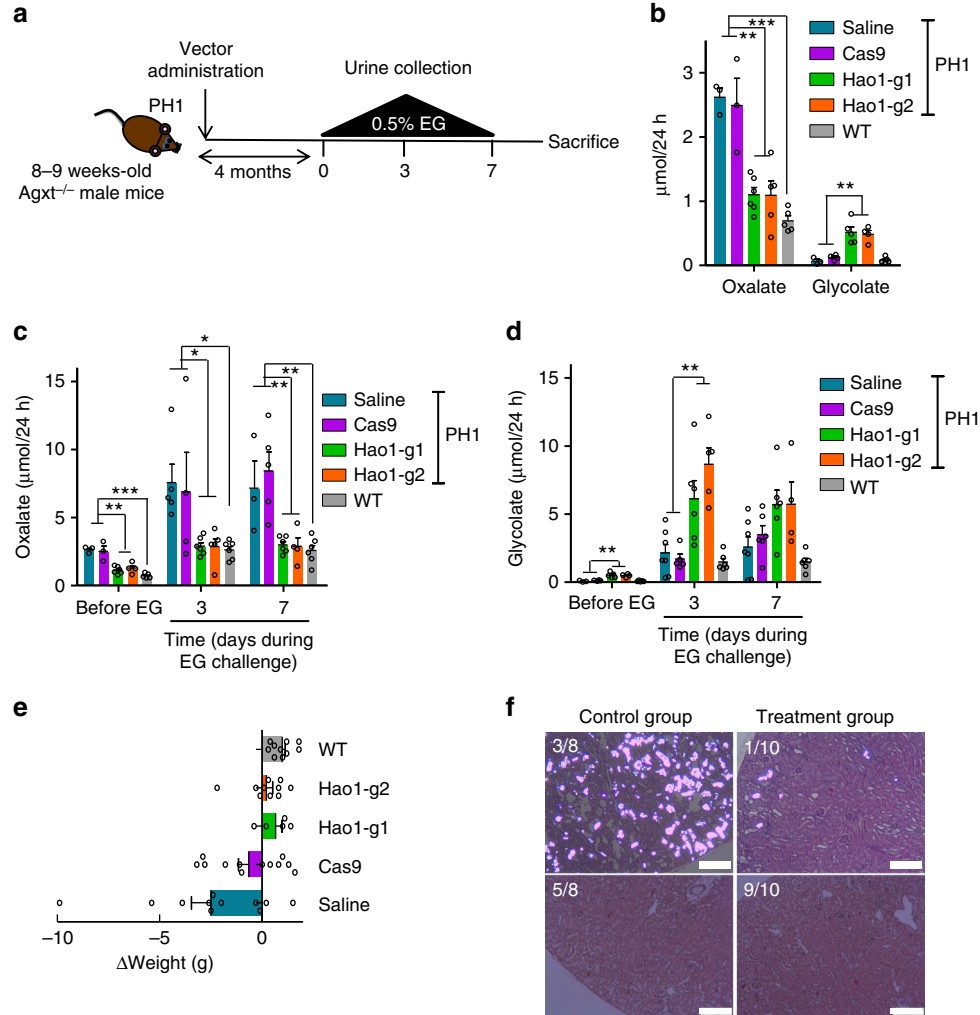

**Fig. 3** Therapeutic efficacy of CRISPR/Cas9-mediated STR in PH1 animals. **a** Schematic experimental procedure, where 8–10-week-old PH1 animals were intravenously treated with saline ($n = 7$), Cas9 ($n = 6$), *Hao1*-g1 ($n = 6$) and *Hao1*-g2 ($n = 5$). A 7-days EG challenge was performed 4 months after vector administration, and 24 h urine samples were collected before and on days 3 and 7 of challenge. **b** Quantification of basal urine levels of oxalate and glycolate (μmol/24 h) 4 months after treatment. **c**, **d** Quantification of urine oxalate (**c**) and glycolate (**d**) levels (μmol/24 h) before and on days 3 and 7 of EG challenge. Data are presented as mean ± SEM and Kruskal–Wallis statistical test was used to compare the groups in each day. **e** Weight change of the animals during a one-week EG challenge. **f** Representative histological analysis of CaOx accumulation in the kidneys of PH1 animals from control or treatment groups after a 10-days EG challenge. Scale bar: 200 μm. *$p < 0.05$; **$p < 0.01$; ***$p < 0.001$

increased indel frequencies at the *Hao1* locus for both sgRNAs ($60.57 \pm 1.80\%$ and $60.13 \pm 5.90\%$, respectively) (Table 1 and Supplementary Fig. 4). These differences are probably due to variability between experiments. Again, the variants introduced were similar to previous observations, with the most common mutations being small indels (<10 bp) triggering a frameshift and with a different indel distribution for each sgRNA (Supplementary Fig. 4). However, none of the frequencies observed at the off-target regions (OFT1 to OFT7) were substantially higher than the background indel rate observed in control animals (usually <0.1% due to sequencing errors) (Table 1). We also analyzed the on-target sites in different tissues, to corroborate the hepato-specificity of our therapeutic vectors. Again, none of the frequencies observed were substantially higher than the control mice (Supplementary Table 3). These results indicate that our therapeutic approach is tissue-specific and there is no in vivo off-target effect in the predicted higher scoring regions for both *Hao1* sgRNAs.

Furthermore, we also analyzed the potential hepatotoxicity of the treatment by studying the histopathology and measuring serum transaminase and bilirubin levels. As shown in Fig. 4a, the livers of treated animals showed normal histology, and CD45 staining of leukocytes revealed no inflammatory infiltrate (quantified in Fig. 4b). The levels of both serum biochemical parameters remained normal in animals receiving the therapeutic vectors (Fig. 4c, d). Moreover, the weight of the animals was monitored weekly during the experiments and normal weight-gain was observed in all the groups before the EG challenge (Supplementary Fig. 5). Although safety in patients cannot be fully predicted, all together, these data suggest that the CRISPR/Cas9-mediated GO inhibition approach to treat PH1 is safe.

## Discussion

The latest advances in CRISPR/Cas9 technology[31,34] have catalyzed the rapid development of new therapies not only for congenital metabolic disease but also for many other inherited disorders, which have shown promising outcomes in animal models[35–43]. In addition to genetic diseases, CRISPR/Cas9 gene editing has also been applied to cancer immunotherapy by engineering autologous T-cells to express chimeric antigen receptors, and to infectious diseases, where genome editing can

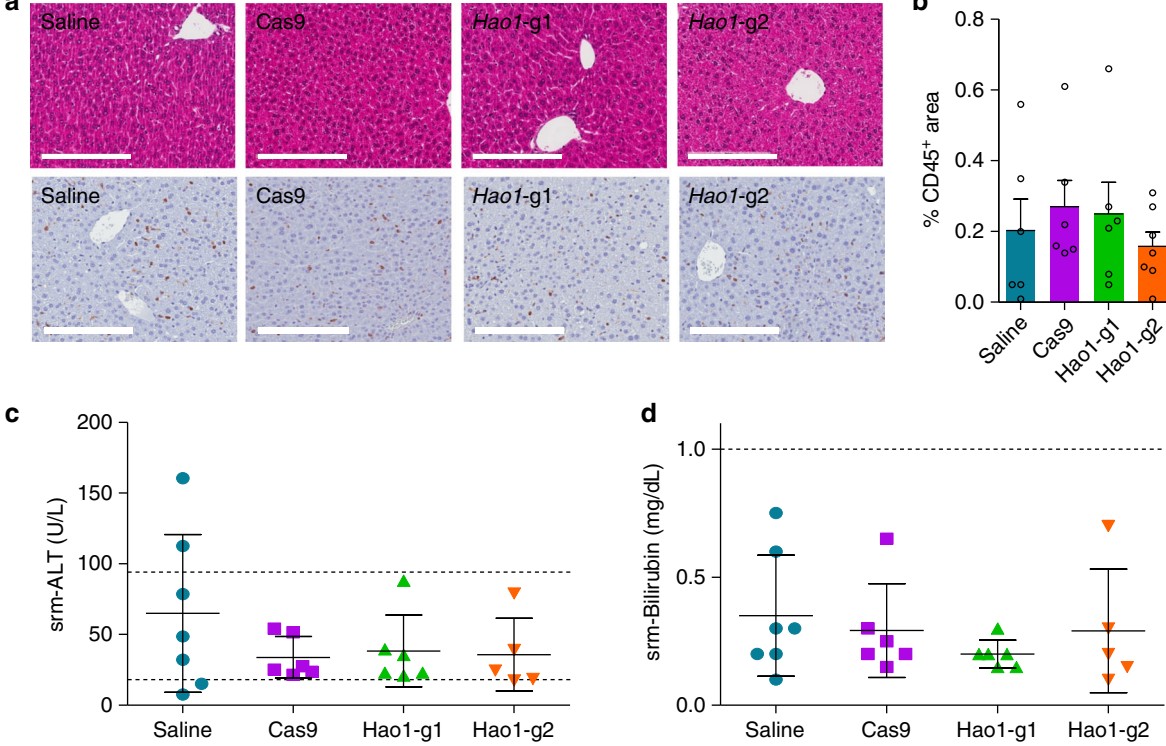

**Fig. 4** Safety of the expression of CRISPR/Cas9 system. **a** Representative hematoxylin-eosin staining and CD45[+] IHC of liver sections from 8–10-week-old PH1 animals sacrificed 4 months after treatment with saline ($n = 7$), Cas9 ($n = 6$), *Hao1*-g1 ($n = 6$) and *Hao1*-g2 ($n = 5$). Scale bar: 200 µm. **b** Quantification of CD45[+] areas (%) of liver sections. (**c**, **d**) Serum ALT (U/L) (**c**) and bilirubin (mg/dL) (**d**) levels measured in animals sacrificed 6 months after treatment. Mean ± SEM of data are presented. Kruskal–Wallis test revealed no significant differences between groups. Dotted lines represent the range of normal reference values for ALT and bilirubin serum levels

modify receptors that are essential for the pathogenesis of viral infections[47].

In this study, we have developed an SRT for primary hyperoxaluria type I (PH1) based on CRISPR/Cas9-mediated gene editing. We have demonstrated that a sustained and highly significant reduction of GO protein expression can be achieved after a single delivery of an AAV vector carrying both *Staphylococcus aureus* Cas9 and GO-specific sgRNAs to the livers of mice with PH1. The high editing efficiency of the system is translated into an outstanding therapeutic effect. Our results clearly demonstrate that renal oxalate accumulation is prevented in treated mice and that they were fully protected against metabolic challenge with an oxalate precursor (EG), showing reduced oxalate excretion in urine and reduced oxalate crystal formation in the kidneys. The detection of indels in treated animals by 3 different methods (Surveyor, TIDE, and NGS) revealed an average of 50–60% of indels, while the GO protein remained undetectable. The reason for the apparent discrepancy between indel frequency and GO disappearance is due to the fact that we performed indel analysis using whole liver samples, which include hepatocytes as well as nonparenchymal cells. Since SaCas9 nuclease expression is restricted to the hepatocytes[44,45], the *Hao1* gene remains intact in the rest of the liver cells. Accordingly, the cleavage efficiency increased when measured in purified hepatocytes, explaining the significant effect observed at protein levels. Similar discrepancies were previously observed when targeting other liver enzymes using CRISPR/Cas9[43,44].

The therapeutic approach developed in this work was safe during the period of the treatment. The only side effect detected when editing *Hao1* was a significant increase of glycolate levels in urine, which is considered non-pathological. Similar results were previously reported in mice deficient in both *Agxt* and *Hao1*[21],

and mice treated with *Hao1* siRNA[22,23]. Furthermore, normal liver histology with no elevation of liver transaminase or bilirubin was observed, indicating that vector administration caused no liver damage. As mentioned, the off-target effect of CRISPR/Cas9 systems is probably the most concerning issue for in vivo therapeutic applications. An important consideration is that a Cas9 nuclease delivered by AAV will be expressed during the whole life of the mice, increasing the probability of non-specific genome modification in the long-term. However, several published studies have showed that undesired mutations attributable to Cas9 are rare[48–50]. Nevertheless, the fact that those studies were performed using zygotes and that the presence of mutations was analyzed in the offspring, make it difficult to compare with a treatment applied to adult mice in a fully mature organ with low cell turnover. In our study, the therapeutic vectors induced liver-specific on-target cleavage, and more interestingly, no off-target effect was observed by NGS at the potential off-target sites in animals treated for at least six months; results that are in accordance with previous studies using similar systems not only in the liver[40–42] but also in other tissues like muscle[36–38]. However, while more experiments are required for an unbiased analysis of non-specific SaCas9 cleavage, there are different alternatives to decrease the probability of off-target modifications, such as the use of the most up-to-date sgRNA design algorithms and selection of guides with low predicted off-target potential[51], or the use of new delivery methods to express or deliver the CRISPR/Cas9 system transiently, which should be sufficient to modify the two *Hao1* alleles. Other alternatives to increase specificity and to eliminate potential off-target effects would be the use of a mutated nickase Cas9 version[52], which only introduces single-strand breaks and requires the use of two sgRNAs, or high-fidelity or self-inactivating Cas9 variants[53,54].

An additional safety concern of the CRISPR/Cas9 system is the immunogenicity of Cas9. Recent findings indicate that cellular and humoral responses against this protein are developed in humans infected with *S. pyogenes* or *S. aureus*[55]. Moreover, AAV-mediated expression of SpCas9 in the muscle of mice also triggered immune responses[56]. However, in our study immunity-related events were not observed in PH1 mice treated with the AAVs expressing SaCas9, since Cas9 expression was not lost and the quantification of the hepatic immune infiltrate did not reveal differences between treated and control groups. Nevertheless, additional analysis should be performed in order to clarify this point and further develop these CRIPSR/Cas9-mediated SRTs as a therapeutic option for metabolic life-threatening diseases that are due to the accumulation of toxic metabolites.

In summary, we have demonstrated that a single administration of an AAV-CRISPR/Cas9 editing vector, targeting one of the enzymes of the glyoxylate metabolic pathway, results in an extraordinary therapeutic effect in PH1 mice in the absence of selective advantage. Our results corroborate the high efficiency of the CRISPR/Cas9 system to knock out the expression of a liver-specific protein in vivo using AAV vectors. Our CRISPR/Cas9-based SRT for PH1 would help to overcome limitations associated with siRNA-mediated approaches that are currently being investigated, avoiding the requirement of multiple administrations for long-term effect[24], which would clearly benefit the quality of life of patients and families. The treatment has been demonstrated to be safe in a PH1 mouse model but more studies in other models need to be performed in order to predict better possible adverse events in patients. Overall, our data provide support to the use of CRIPSR/Cas9-mediated GO depletion as an effective and alternative promising therapeutic strategy for PH1 patients. Furthermore, CRISPR/Cas9-based SRTs could be applied to PH2 and PH3 by targeting other enzymes involved in glyoxylate metabolism, such as lactate dehydrogenase (LDH) or hydroxyproline dehydrogenase (HYPDH or PRODH2). In particular, liver-specific suppression of *Ldha* mRNA expression by siRNA efficiently reduces urinary oxalate levels in all tested hyperoxaluria animal models[57]. Moreover, CRISPR/Cas9-based SRTs could also be applied to other inherited metabolic diseases, such as lysosomal storage disorders (LSD) in particular. For example, inhibition of *Gly1* has been proposed as a therapeutic approach for Pompe disease[58], while *Ext1* and *Ext2* inhibition reduces disease-specific biomarkers and the effects caused by heparan sulfate accumulation in Mucopolysaccharidoses type III[59]. Thus, CRISPR/Cas9-mediated inhibition of these genes could offer a potential therapeutic approach for these devastating diseases.

## Methods

**CRISPR/Cas9 vectors and gRNA design.** The pX602-AAV-TBG::NLS-SaCas9-NLS-HA-OLLAS-bGHpA;U6::BsaI-sgRNA plasmid that contains the *Staphylococcus aureus* Cas9 (SaCas9) expressed under the tTBG promoter, the sgRNA under U6 promoter and ITR sequences for AAV vector production was a gift from Feng Zhang (Addgene plasmid # 61593)[44]. sgRNAs targeting exonic regions of the *Hao1* gene were designed and selected using Benchling software (www.benchling.com). The selected 21-nt sequences upstream of the 5′-NNGRRT-3′ PAM sequence of SaCas9 are shown in Supplementary Table 1. Annealed oligonucleotides (Sigma) were cloned into BsaI site of pX602 vector using standard molecular cloning techniques.

**AAV8 vector production.** Serotype 8 AAV vectors containing each *Hao1* sgRNA were produced in HEK293T (ATCC CRL-3216) packaging cells as described[60]. Briefly, for each production, the shuttle vector for AAV and the packaging plasmid pDP8.ape (Plasmid factory) were co-transfected into HEK293T cells. The cells and supernatants were harvested 72 h upon transfection and virus was released from the cells by three rounds of freeze–thawing. Crude lysate from all batches was then treated with DNAse and RNAse (0.1 mg per p150 culture dish) for 1 h at 37 °C and then kept at −80 °C until purification. Purification of crude lysate was performed by ultracentrifugation in Optiprep Density Gradient Medium-Iodixanol (Sigma-Aldrich). Thereafter, iodioxanol was removed and the batches concentrated by passage through Amicon Ultra-15 tubes (Ultracel-100K; Merck Millipore). AAV8

vector without sgRNA (Cas9) was produced as control. For virus titration, viral DNA was isolated using the High Pure Viral Nucleic Acid kit (Roche Applied Science). Viral titers in terms of viral genome per milliliter (vg/mL) were determined by qPCR (Applied Biosystems) using SaCas9 specific primers (Supplementary Table 4).

**Animal procedures.** All experimental procedures were approved by the Ethics Committee of the University of Navarra and the Institute of Public Health of Navarra according to European Council Guidelines. All the experiments using animal models complied with all relevant ethical regulations. *Agxt1*$^{-/-}$ mice (B6.129SvAgxt$^{tm1Ull}$) mice were bred and maintained in a pathogen-free facility with free access to standard chow and water. *Agxt1*$^{-/-}$ mice were genotyped as described[16], by using KAPA HotStart Mouse Genotyping Kit (Kapa Biosystems) with wild-type and mutant-specific primers together with a common forward primer (Supplementary Table 4). Age-matched C57BL/6J mice (Harlan laboratories) were used as control animals. $5 \times 10^{12}$ vg/kg were administered to 12–14-week-old *Agxt1*$^{-/-}$ male animals (in short-term studies) or 8–10-week-old *Agxt1*$^{-/-}$ animals (long-term studies) by intravenous injection. Animals were continuously challenged with 0.5% (v/v) EG in drinking water for 7–10 days. During the challenge, mice were individualized in metabolic cages for the collection of 24 h urine and monitoring of water intake. Urine samples were obtained before, and on days 3 and 7 during the challenge. Serum samples were obtained before the euthanasia and liver and kidney samples during the necropsy.

**Genomic DNA extraction and indel quantification.** Genomic DNA was extracted from tissue sections using NucleoSpin Tissue DNA extraction kit (Macherey-Nagel) according to the manufacturer's instructions. The efficiency of each individual sgRNA was tested by surveyor using the Surveyor® Mutation Detection Kit (IDT) following the manufacturer's instructions. Uncropped electrophoresis gels are available as Supplementary Information (Supplementary Fig. 6). Indel percentage was calculated by sequencing the amplification products of *Hao1* gene exon 2 and comparing the chromatograms with control samples using TIDE webtool (https://tide.nki.nl)[46].

**On-target and off-target mutagenesis analyses.** To further analyze off-target sites, the most likely off-target sites for each selected sgRNA were determined using Benchling software (www.benchling.com) (Supplementary Table 2). A total of 7 off-target candidates were selected for each sgRNA. Primers spanning on-target and off-target sites for each sgRNA (Supplementary Table 4) were used to amplify relevant sequences by nested PCR. Briefly, 50 ng of genomic DNA was first amplified using specific primers using KAPA HiFi HotStart ReadyMix PCR Kit (Kapa Biosystems) and purified using Agencourt AMPure XP system (Beckman Coulter). Then, for each sgRNA, equal amounts of purified amplicons from the same sample were mixed. Nested PCR was performed using 10 ng of PCR mix and universal primers with specific barcodes for each sample were used to generate Illumina amplicons. PCR products were purified as described above. Final library was made mixing equal amounts of the second PCR products and sequenced on Illumina MiSeq ($2 \times 250$ bp paired-end) at >25,000× coverage at amplified regions. Data were processed according to standard Illumina sequencing analysis procedures. Processed reads were aligned to the reference (GRCm38/mm10) using Burrows-Wheeler Aligner (BWA). Reads that did not map to the reference were discarded. Insertions and/or deletions were determined by comparison of reads against reference using CrispRVariants R-based toolkit[61].

**Liver transduction and gene expression by RT-qPCR.** Genomic DNA was extracted as explained above and qPCR was employed to quantify viral genome copies in the liver using SaCas9 specific primers (Supplementary Table 4). RNA was isolated using Trizol (Life Technology) and reverse-transcribed using Prime-Script™ RT reagent Kit (Takara). qPCR to quantify *Hao1*, SaCas9 and *β-actin* were performed in a QuantStudio™ 5 Real-Time PCR System (ThermoFisher Scientific) or a CFX96™ Real-Time PCR Detection System (Bio-Rad) using gene-specific primers (Supplementary Table 4). Data were normalized to *β-actin* levels.

**GO detection by western blot and immunohistochemistry.** For the detection of GO by western blot, liver total proteins were extracted using RIPA Buffer and quantified using BCA Protein Assay Kit (ThermoFisher Scientific). Proteins were separated using a 15% acrylamide gel and blotted in a nitrocellulose membrane. Membranes were blocked using SuperBlock™ (TBS) Blocking Buffer (ThermoFisher Scientific) and incubated with anti-GO antibody (rabbit serum raised against recombinant mouse GO protein) diluted 1:2500 overnight (o/n) at 4 °C in Blocking Solution. The membranes were washed using TBS-0.05% Tween solution. Anti-GAPDH antibody (#G8795, Sigma-Aldrich) was used at 1:5000 dilution during 2 h at RT as internal reference. SuperSignal West Femto Maximum Sensitivity Substrate (ThermoFisher Scientific) was used for the development of the signal after incubation with the corresponding secondary antibody (HRP-conjugated anti-rabbit IgG #NA934V from GE Healthcare at 1:2500 dilution or HRP-conjugated anti-mouse IgG #NA931V from GE Healthcare at 1:5000 dilution) during 1 h at RT. Odyssey Fc (LI-COR) imaging system was used for image generation. Uncropped blots are available as Supplementary Information (Supplementary

Fig. 7). Formalin-fixed paraffin-embedded liver sections were used for immuno-histochemical detection of GO using Benchmark Ultra equipment and reagents (Ventana). Antigen retrieval was performed at 95 °C in high pH buffer (CC1, Ventana) for 24 min, and followed by incubation with anti-GO rabbit serum diluted 1:500 during 28 min at 37 °C. Next, polymeric HRP-conjugated anti-rabbit IgG secondary antibody (#760-4311, Ventana Discovery Systems) at 1:1000 dilution was applied and a chromogenic signal was developed with DAB and copper enhancement (Ventana).

**Urine oxalate/glycolate measurement and serum biochemistry.** Individual 24 h urines were collected in 50 µl of 5 N HCl and their volume was measured. The urines were treated with activated charcoal and urine oxalate levels were measured using the Oxalate kit (Trinity Biotech) according to the manufacturer's instructions. Oxalate quantification (µmol/24 h) was calculated using the oxalate concentration and the urine volume. Glycolate concentration in charcoal-treated urine samples was measured in a colorimetric assay using recombinant GO enzyme in 50 mM potassium phosphate buffer pH 7, coupled with HRP in the presence of sulfonated-DCIP and 4-aminoantipyrine (Sigma Aldrich) as chromogens. The absorbance was measured relative to a glycolate standard curve 515 nm[62]. Peripheral blood samples were collected at the indicated time points, and the serum was separated from whole blood after centrifugation at 845xg for 15 minutes. Serum alanine aminotransaminase (ALT) and bilirubin levels were quantified using a Cobas C111 analyzer (Roche).

**Histological analysis of kidney.** Tissues were fixed in 4% buffered paraformaldehyde and either embedded in paraffin or cryoprotected in 20% sucrose and snap frozen in liquid nitrogen. Routine hematoxylin-eosin stained sections were analyzed under dark field optics to count typical birefringent oxalate deposits. Three cortical areas were photographed with a ×5 objective lens in a microscope (Leica DMLB) fitted with polarization filters to generate a dark field in which calcium oxalate crystals are seen as bright object. Digitized images were treated with ImageJ software to calculate the percentage area above a threshold established with a normal kidney slide photographed with the same settings. Four ranks of nephrocalcinosis were stablished depending on crystal quantification: negative (0%), mild (0–0.99%), moderate (1–5%), and severe (>5% area occupied by calcium oxalate crystals).

**Hematoxylin-eosin staining and CD45⁺ IHC in liver sections.** Liver was harvested and fixed in 4% formalin. Paraffin liver sections (3 µm thick) were cut, dewaxed and hydrated. Hematoxylin-eosin staining was performed for structural analysis. For IHC, antigen retrieval was applied: 2 µg/ml proteinase K (Sigma) for 30 min at 37 °C or heating for 30 min at 95 °C in 0.01 M Tris-1 mM EDTA pH 9 in a Pascal pressure chamber (Dako). Endogenous peroxidase was blocked with 3% $H_2O_2$ in deionized water for 10 min and sections were washed in TBS-0.05% Tween 20. Incubation with rat anti-CD45 primary antibody (#103102, Biolegend) at 1:2000 dilution was performed overnight at 4 °C. After rinsing in TBS-T, the sections were incubated with the rabbit anti-rat IgG secondary antibody (#E0468, Dako) for 30 min at RT followed by incubation with EnVision anti-rabbit (#K401, Dako) for other 30 min at RT. Peroxidase activity was revealed using DAB+ and sections were lightly counterstained with Harris hematoxylin. Finally, slides were dehydrated in graded series of ethanol, cleared in xylene and mounted with Eukitt (Labolan).

**Statistics.** All the results are presented as mean ± standard error of the mean (SEM) of at least five mice per group at each time point to ensure reproducibility. Sample sizes are noted in figure legends. Graphs and statistical analyses were performed using GraphPad Prism 7.0d. Kruskal–Wallis test with Dunn's post-test was used to analyze differences between groups. Friedman test with Signs post-test was used for comparison between different administration groups at different times. Mann–Whitney test was used for individual comparisons. $p$-values < 0.05 were considered significant.

## Data availability
The authors declare that the data supporting the findings of this study are available within the article, the Supplementary Information files or are available in a public database. Sequencing raw data (fastq files) have been deposited in the SRA database with the SRP159060 accession code and will be available upon publication.

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

## Acknowledgements

We thank the Bioinformatics Unit at CIMA and the Genomics Facility at CIMA LAB Diagnostics. This work was funded by grants from ERA-NET E-Rare 3 research program JTC ERAdicatPH, Instituto de Salud Carlos III (ISCIII) AC15/00036 and PI16/00150, TERCEL (ISCIII) RD16/0011/0005, MINECO SAF2015-69796, Gobierno de Navarra 91/2016, Ayuda a la investigación D. Juan Manuel Mingo and Oxalosis & Hyperoxaluria Foundation. NeZ was supported by fellowships form Asociación Amigos de la Universidad de Navarra (ADA) and Fundación para la Investigación Médica Aplicada (FIMA).

## Author contributions

Conception and Design: Ne.Z., E.S., F.P., G.G.A. and J.R.R.M. Development of methodology: A.V.Z., L.C. and D.L.A. Acquisition of data: Ne.Z., M.B., C.M.H., Na.Z., I.B., S.R., R.M.T., L.T., A.V., C.O., E.S. and J.R.R.M. Analysis and interpretation of data: Ne.Z., E.S., G.G.A. and J.R.R.M. Writing and/or revision of the manuscript: Ne.Z., E.S., F.P., D.L.A., G.G.A. and J.R.R.M. Study supervision: E.S., F.P., G.G.A. and J.R.R.M. All figures and graphs depicted in this manuscript were created by the authors of this work.

## Additional information

**Competing interests:** Eduardo Salido holds shares of Orfan Biotech. Gloria Gonzalez-Aseguinolaza is a founder and shareholder of Vivet Therapeutics. The remaining authors declare no competing interests.

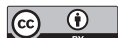

