## [Peer Review File · Nature Communications]

Reviewer #1 - CRISPR/Cas9 using AAV in liver (Remarks to the Author):

Zabaleta et al. describes a therapeutic gene editing approach using CRISPR/Cas9 for primary hyperoxaluria type I. Using CRISPR/Cas9 in vivo, they delete a gene (Hao1) in a disease-associated pathway to reduce the substrate of the metabolic pathway and render the phenotype benign. Such a substrate reduction therapy using genome engineering has been defined previously as metabolic pathway reprogramming (Pankowitz et al. Nat Commun 2016) and reduces the conceptual novelty of this manuscript. Nevertheless, HP1 has indeed only poor therapeutic alternatives and a novel therapeutic approach would be desirable. While the manuscript is generally well written and the figures are clearly presented, a major concern is the therapeutic read out. There is no phenotypical readout and the biochemical parameters are not complete nor convincingly different in the treatment group (Fib. 3b and 3c: decreased oxalate and increased glycolate after EG). While the former is difficult to overcome, the second needs to be addressed in a convincing manner.

Major points:

Experimental read-out:

- 1) Experimental endpoints: plasma levels oxalate missing, urine levels unlikely to be significant different using proper statistical analysis. I doubt that overlapping SEM of groups to compare are statistically different (see below). More mice need to be analyzed to show statistical significance.
- 2) Statistical analysis: WT and Hao1-g1 and Hao1-g2 cannot be taken together to compare to saline and Cas9.
- 3) Instead of a one-time challenge, a constant challenge would increase the validity and confirm that the treatment trend is indeed a statistical significant difference.

Minor Concerns:

1. Figure 3b and 3c x-axis should read as "post-challenge" since it is stated that the EG is administered only once and the oxalate/glycolate levels are measured 3 and 7 days after the EG challenge.
2. The age of these mice in figure 2 are never explicitly stated in either the figure legend or the manuscript's text.
3. In Table 1, why does the %ONT average increase after 6 months? There were no growth or selection advantages described, so please explain these findings.

4. Supplemental figure 1 should include the animal's age in the figure legend.
5. Figure 3e is missing scale bars and should include quantifications. Overview pictures are needed.
6. What percentage of patients progress to renal failure or end-stage renal disease? The authors imply that PH1 is a more dramatic disease, but the majority of compliant patients are fine and would not want to undergo any therapy...
7. Discrepancy of mRNA and protein levels does not reflect previous literature of the two cited papers. Explain scientifically (protein stability, sampling error, etc.).

Reviewer #2 - Hyperoxaluria (Remarks to the Author):

A very nice study that demonstrates the potential effectiveness of this approach in an inbred knockout mouse model.

Comment 1 Introduction

Please provide reference for the statement that 80% of cases are PH1. Recent studies suggest somewhat lower prevalence when the non PH1 /2/3 are considered.

Comment 2 Introduction

Please make it clear that liver transplant is only an established therapy for PH1.

Comment 3 Introduction

Please clarify what are the potential "drug interactions" for an siRNA approach.

Comment 4 Introduction

Although the effects of the CRISPR/Cas 9 seem long term, it's not clear one can claim it's permanent.

Comment 5 Results

Please provide some semi quantitative or quantitative analysis of the renal histology.

Comment 6 Results (last sentence)

The current study does not prove this is a safe approach in humans. Indeed, since humans (unlike mice) are not inbred, the potential for unintended off target effects is possibly much greater (and perhaps more sporadic from person to person). Given the long term effects of this treatment, indeed, this could be a great disadvantage of this approach.

Comment 7 Discussion

It would seem enough to point out the differences between studies to date, rather than using terms such as “extensively criticized by a number of scientists”, especially since this therapy has yet to be tried in humans.

Comment 7 Discussion (last sentence)

The final sentence is much too definitive since the current study does not prove that this strategy is definitive or safe for human patients with PH1.

Comment 8 (overall)

Please be clear throughout that this strategy (targeting GO) is likely to only be effective for PH1 (as opposed to PH2 and 3). This is implied perhaps, but not stated.

Reviewer #3 - Hyperoxaluria therapy (Remarks to the Author):

Substrate reduction has previously been shown to be an effective therapy for the treatment in a mouse model of the monogenic disease, Primary Hyperoxaluria Type 1 and more recently in patients using siRNA approaches to diminish the amount of glyoxylate produced from glycolate oxidation. In this manuscript the authors describe the utility of modifying the genomic sequence of the liver-specific enzyme, glycolate oxidase (GO), to reduce oxalate production in this mouse model to near normal. CRISPR/Cas9 technology was used coupled with an AAV vector for hepatic delivery. This research suggests that this approach could potentially be used therapeutically to treat this disease in

humans. Further refinements appear to be warranted before this treatment is ready for clinical trials, however.

The effect of the treatment did not produce a change in the production of GO mRNA (Fig. 1c). The reasons for the large variability between mice in each group should be explained.

There was also a variable expression of GO protein in both saline and Cas9-treated mice in Western blots with no explanation (Fig. 1d and Suppl 1e).

Although oxalate excretion is significantly reduced by about half in treated Agxt mice (Fig. 3b), it appears to be twice that in WT mice. A clearer presentation of the results in a larger figure may clarify this picture which is the most relevant endpoint in their studies.

The Indel frequency suggests that genomic alterations occurred in only half of the hepatocytes with both guide RNAs. Determining correlations between Indel frequencies and oxalate excretion in each mouse may show how this relationship effects the treatment.

If only half of the hepatocytes are modified this would appear to be insufficient to return oxalate production to normal.

Cas9 treatment (Fig. 1b) suggests that it causes a low number of Indels in 2 of 5 mice. Is this “noise” or to be expected? No changes were detected in Fig. 2.

Fig. 2 is duplicated in the Suppl figures as Fig. 4.

The frequency of Hao1 deficiency in the population is not known, possibly because of its mild metabolic effects. Because of the paucity of individuals studied it is premature, however, to conclude the consequences are benign as stated on L87.

The large variations in ALT and Bilirubin in Fig. 4b indicate that in this model these assays are not reliable indicators of hepatotoxicity. H&E stains of liver sections would also be helpful to indicate the presence or absence of inflammation.

The inconsistencies in the data identified above require more explanation and need to be identified as limitations.

Response to the Reviewers' comments

(manuscript NCOMMS-18-02083 by Zabaleta *et al.*)

We would like to thank the reviewers for their thorough revision and constructive criticisms and suggestions on our manuscript. We have made every effort to take them into consideration when correcting the text. We believe that this revised version of our study has clearly improved following the reviewers' advice. Below you can find the detailed answers to all the issues raised by the reviewers.

Reviewer #1 - CRISPR/Cas9 using AAV in liver (Remarks to the Author):

Zabaleta et al. describes a therapeutic gene editing approach using CRISPR/Cas9 for primary hyperoxaluria type I. Using CRISPR/Cas9 in vivo, they delete a gene (Hao1) in a disease-associated pathway to reduce the substrate of the metabolic pathway and render the phenotype benign. Such a substrate reduction therapy using genome engineering has been defined previously as metabolic pathway reprogramming (Pankowitz et al. Nat Commun 2016) and reduces the conceptual novelty of this manuscript. Nevertheless, HP1 has indeed only poor therapeutic alternatives and a novel therapeutic approach would be desirable. While the manuscript is generally well written, and the figures are clearly presented, a major concern is the therapeutic read out. There is no phenotypical readout and the biochemical parameters are not complete nor convincingly different in the treatment group (Fib. 3b and 3c: decreased oxalate and increased glycolate after EG). While the former is difficult to overcome, the second needs to be addressed in a convincing manner.

Major points:

Experimental read-out:

1) Experimental endpoints: plasma levels oxalate missing, urine levels unlikely to be significant different using proper statistical analysis. I doubt that overlapping SEM of groups to compare are statistically different (see below). More mice need to be analyzed to show statistical significance.

The reviewer is absolutely right, we apologize since we have made a mistake in the representation of the data. We represented in Figure 3b (now Figure 3c) the oxalate values as the mean \pm SD instead of mean \pm SEM as we indicated in figure legend. In the corrected version of the figures, we have represented the mean \pm SEM and performed a correct statistical analysis. The statistical test used to compare the mean of each group was Kruskal Wallis and as post-hoc Dunn's test was applied. *p* values between all groups were 0.0048 (**), 0.0223 (*) and 0.0054 (**) for days 0, 3 and 7 of challenge, respectively. Dunn's test revealed significant differences in oxalate levels between control and treated groups as well as between control and WT groups at all time points (depicted in the Figure 3c). We made the same mistake in the

graph representing glycolate values (now Figure 3d). We have corrected both mistakes in the revised version of the manuscript.

We agree with the reviewer that the measurement of serum oxalate would provide valuable information about the metabolic correction in PH1 mice. However, the method to measure oxalate levels (enzymatic assay) is not sensitive enough to detect the low amounts of oxalate present in a mouse serum samples. It is also known that serum oxalate levels only increase after renal function decline, particularly in mice, since they normally have much higher glomerular filtration rate than humans. After severe renal damage, serum oxalate is expected to increase but the time window to detect this increase in mice is very narrow and animals die suddenly once renal function declines. These technical considerations made it difficult for us to record serum oxalate levels.

On the other hand, since urine oxalate levels in the PH1 animals from the untreated control groups were very variable, additional animals were included in the different groups following the referee's suggestion. However, the 7-days EG challenge (0.5% in drinking water during 7 consecutive days) was performed 1 month after the administration of the treatment. We enclose the results for your consideration (Figure A). As can be seen in the Figure A, we also observed a significant reduction in 24h urine oxalate levels in the animals treated with the therapeutic vectors compared to controls during the EG challenge. We only observed a slight difference with the data presented in the original manuscript (Figure 3b, now Figures 3b and 3c) in the basal oxalate levels before EG challenge, that in this case, were not normalized one month after treatment. We can only speculate about the reason for this finding that might be related to the difference in the timing between treatment and the analysis of the basal oxalate values. Suggesting the requirement of a readjustment of the glyoxylate metabolic flux after glycolate oxidase deletion.

Figure A. Therapeutic efficacy of CRISPR/Cas9-mediated STR in PH1 animals 1 month after treatment. Quantification of urine oxalate levels ($\mu\text{mol}/24\text{h}$) before and on days 3 and 7 of 0.5% EG challenge in animals treated with saline ($n \geq 8$), Cas9 ($n \geq 7$), *Hao1-g1* ($n \geq 6$) and *Hao1-g2* ($n \geq 7$). A group of wild type (WT) animals ($n \geq 8$) was also included. Kruskal Wallis statistical test was used to compare the groups in each day. * $p < 0.05$; ** $p < 0.01$; *** $p < 0.001$

2) *Statistical analysis: WT and Hao1-g1 and Hao1-g2 cannot be taken together to compare to saline and Cas9.*

We apologize for the mistake in the representation of the data in the Figure 3b (now Figure 3c). We have analyzed the difference between the treated, untreated and the WT groups independently and the data have been properly represented.

3) *Instead of a one-time challenge, a constant challenge would increase the validity and confirm that the treatment trend is indeed a statistical significant difference.*

We apologize for not explaining properly the protocol of challenge with EG. The animals received EG in drinking water for 7 consecutive days, but the oxalate in urine was analyzed before challenge and at days 3 and 7 of challenge, a better explanation was introduced in the text. We agree with the reviewer that a more prolonged challenge would be ideal for the study of therapeutic efficacy of the vector, which would resemble the condition of PH1 patients. However, when we performed a continuous 15-day EG challenge, PH1 animals were not able to survive, and by day 13 we had problems in urine collection due to the significant deterioration of the health status of the PH1 animals (Figure B). We show these results for reviewer's consideration.

Figure B. Quantification of urine oxalate levels ($\mu\text{mol}/24\text{h}$) in WT and PH1 animals during a continuous EG challenge in drinking water (0.5%).

In order to further validate that our CRISPR/Cas9-mediated SRT treatment was effective in more severe challenge conditions and due to the impossibility to perform a more prolonged challenge, we decided to perform a double challenge experiment. Thus, PH1 animals were treated with saline, Cas9 or *Hao1-g2* and 1 month after vector administration a 7-day EG challenge was performed. After 15 days of resting, same animal groups were re-challenged for another 7 days with EG. We observed a significant reduction in urine oxalate ($\mu\text{mol}/24\text{h}$) in the animals treated with the therapeutic vector (*Hao1-g2*) compared to controls (Figure C). These results indicate the robust efficiency of our CRISPR/Cas9-mediated SRT treatment after two consecutive challenges.

Figure C. Therapeutic efficacy of CRISPR/Cas9-mediated STR in PH1 animals after double challenge. Quantification of urine oxalate levels ($\mu\text{mol}/24\text{h}$) before and on days 3 and 7 of each 0.5% EG challenge in PH1 animals treated with saline, Cas9 and Hao1-g2. A group of wild type (WT) animals was also included. Kruskal Wallis statistical test was used to compare the groups in each day.

Minor Concerns:

1. Figure 3b and 3c x-axis should read as “post-challenge” since it is stated that the EG is administered only once and the oxalate/glycolate levels are measured 3 and 7 days after the EG challenge.

As explained in the previous answer to the reviewer comment, we did not properly describe the protocol of challenge since we have administered a continuous EG in drinking water during 7 days and not just once. We have modified the legend of the x-axis in Figures 3c and 3d of the revised version of the manuscript to make it clearer.

2. The age of these mice in figure 2 are never explicitly stated in either the figure legend or the manuscript’s text.

We have included the age of the animals in this experiment in *Material and Methods* section (L357-358) as well as in the figure legends.

3. In Table 1, why does the %ONT average increase after 6 months? There were no growth or selection advantages described, so please explain these findings.

According to the data showed in table 1 the percentage of indels increased when measured after 6 months, from $48.46 \pm 3.41\%$ to $60.57 \pm 1.79\%$ for *Hao1-g1*, and from $52.81 \pm 3.89\%$ to $60.13 \pm 5.90\%$ for *Hao1-g2*. However, those differences are only significant for *Hao1* gRNA1 ($p=0.0079$, Mann Whitney test). We reasoned that these differences could be due to: i) the experimental variability between treated mice; ii) the different age of the animals at the time of injection (12-14 weeks for the analysis after 1 month, and 8-10 weeks for the analysis after 6 months); or iii) that cleavage efficiency increases with time due to the continuous expression of Cas9 and gRNAs.

In order to address these questions, we performed several experiments:

i) First, to check if the experimental variability could explain differences in the observed cleavage efficiencies, we repeated the analysis of 1 month after vector injection and analyzed the indel frequency (% of variants) by NGS in 4 new 12-14 week-old PH1 animals injected with the same dose (5×10^{12} vg/kg) of *Hao1-g1* therapeutic vector. We observed an average of $55.05 \pm 2.16\%$ of indels, that is statistically higher ($p=0.0317$, Mann Whitney test), compared to the first analyzed group included in the manuscript. These results clearly indicate that there is an experimental variability when measuring cleavage efficiency.

ii) Second, to elucidate the effect of the age of the animals at the time of treatment, we compared the indel frequency 1 month after *Hao1-g1* vector injection (5×10^{12} vg/kg) in 12-14 week-old versus 8-10 week-old PH1 animals. We did not observe significant differences between the groups ($55.05 \pm 2.16\%$ vs $47.98 \pm 6.16\%$ respectively, $p=0.2$, Mann Whitney test), indicating that the age of the PH1 animals was not affecting the cleavage efficiency.

iii) Finally, regarding a time dependent effect on DNA cleavage, as the reviewer indicates, there is no selective advantage for corrected hepatocytes in PH1, but it could be related to the continuous expression of Cas9 and the guide. Several reports have demonstrated that CRISPR/Cas9 cleavage *in vitro* is time and dose dependent (Cameron et al., 2017; Tsai et al., 2017). We can elucidate that the long-term Cas9/gRNA expression from the AAV vector could favor the cleavage of untargeted alleles; however, this would be difficult to address since we would need to analyze the cleavage efficiency on the same mice at different time points. Moreover, according to the literature Cas9-mediated cleavage takes place during the first hours/days after Cas9-gRNA expression. Thus, the slight differences observed between 1 and 6 months in table 1, together with the experimental variability described above, would suggest that the differences observed in our study would be more related to the experimental variability than to a long-term Cas9 expression.

4. Supplemental figure 1 should include the animal's age in the figure legend.

According to the reviewer's suggestion we have included the age of the animals in figure legend.

5. Figure 3e is missing scale bars and should include quantifications. Overview pictures are needed.

We apologize for this mistake and we have included scale bars in Figure 3e (now Figure 3f), that has been also extended in order to represent better the results obtained in this experiment. Thus, three cortical areas were photographed with a 5x objective lens in a microscope (Leica DMLB) fitted with polarization filters to generate a dark field in which calcium oxalate crystals are seen as bright object (new Supplementary figure 3b). Digitized images were treated with ImageJ software to calculate the percentage area above a threshold established with a normal kidney slide photographed with the same settings. Supplementary figure 3b shows representative images from the four ranks of nephrocalcinosis found: negative (0%), mild (0-0.99%), moderate (1-5%), and severe (>5% area occupied by calcium oxalate crystals).

6. What percentage of patients progress to renal failure or end-stage renal disease? The authors imply that PH1 is a more dramatic disease, but the majority of compliant patients are fine and would not want to undergo any therapy...

Primary Hyperoxaluria type 1 is the most common and severe form of the inborn errors of metabolism causing hyperoxaluria by mutations in AGXT gene, resulting in progressive decrease of glomerular filtration rate (GFR), and ultimately leading to end stage renal disease (ESRD) and, if untreated, death in most patients (Leumann and Hoppe, 2001). There is wide phenotypic variability in PH1. Some patients develop severe infantile oxalosis, with ESRD in the first few years of life, while others reach ESRD in the thirties (Leumann and Hoppe, 2001; van Woerden et al., 2003). The natural history of untreated PH1 is one of progressive decline in renal function as a result of calcium oxalate deposits in kidney tissue and complications of nephrolithiasis such as obstruction and infection. A recent study found renal survival in individuals with PH1 to be 76%, 43%, and 12% at ages 20, 40, and 60 years, respectively (Hopp et al., 2015). When oxalate excretion in the urine is insufficient to prevent systemic build-up of oxalate, systemic deposition of calcium oxalate (oxalosis) becomes life threatening due to cardiac arrhythmias and A-V block, cardiomyopathy, vasculopathy, bone and joint disease, refractory anaemia and retinopathy (Cochat and Rumsby, 2013). Early diagnosis and initiation of conservative therapy are critical in preserving adequate renal function for as long as possible (Fargue et al., 2009). About a third of PH1 patients (mostly those homozygous for the mistargeting mutation p.Gly170Arg) respond to treatment with pyridoxine (vitamin B6, precursor to PLP) and can preserve renal function for decades (Hoyer-Kuhn et al., 2014). Once GFR <25-30 mL/min/1.73 m² combined liver-kidney transplantation is the preferred option for treatment (Cochat et al., 2012). The percentage of survivors after liver-kidney transplant is better than after isolated kidney transplant (Bergstralh et al., 2010; Harambat et al., 2012). Thus, there is a significant percentage of PH1 patients that will benefit from advanced genetic therapies, particularly those with severe infantile oxalosis.

7. Discrepancy of mRNA and protein levels does not reflect previous literature of the two cited papers. Explain scientifically (protein stability, sampling error, etc.).

We agree with the reviewer about the discrepancy between the mRNA levels in our study and previous published data. We hypothesized that this might be due to the large variability of the data, which can be associated with the fact that *Hao1* mRNA levels were analyzed in the livers of animals that had been challenged with EG just before sacrifice. We have repeated the experiment but sacrificing the animals in the absence of EG challenge, and we were able to observe a decrease in mRNA levels in the animals from *Hao1-g1* and *Hao1-g2* treated groups. We have changed the Figure 1c and introduced the correct data in the revised version of the manuscript, and the text has been changed in accordance (L154-155). These data indicate that the administration of EG led to changes in *Hao1* expression in all the animals, because we observed increased *Hao1* mRNA levels in untreated PH1 mice after an EG challenge.

Reviewer #2 - Hyperoxaluria (Remarks to the Author):

A very nice study that demonstrates the potential effectiveness of this approach in an inbred knockout mouse model.

Comment 1 Introduction

Please provide reference for the statement that 80% of cases are PH1. Recent studies suggest somewhat lower special when the non PH1 /2/3 are considered.

It is likely that as more patients are diagnosed with the recently identified PH type 3 (PH3), the percentage of PH1 among primary hyperoxalurias will go down. We used 80% as the percentage quoted in reviews from large hyperoxaluria centers (Edvardsson et al., 2013; Hopp et al., 2015). However, we agree with the reviewer that it would be safer to say that about 70-80% of cases are PH1, while we wait for more current statistics on the relative percentages of the various types of PH. This aspect has been modified in the revised version of the manuscript (L72).

Comment 2 Introduction

Please make it clear that liver transplant is only an established therapy for PH1.

Early diagnosis and initiation of conservative therapy are critical in preserving adequate renal function for as long as possible (Fargue et al., 2009). About a third of PH1 patients (mostly those homozygous for the mistargeting mutation p.Gly170Arg) respond to treatment with pyridoxine (vitamin B6, precursor to PLP) and can preserve renal function for decades (Hoyer-Kuhn et al., 2014). Once GFR <25-30 mL/min/1.73 m² combined liver-kidney transplantation is the preferred option for treatment (Cochat et al., 2012). The percentage of survivors after liver-kidney transplant is better than after isolated kidney transplant (Bergstralh et al., 2010; Harambat et al., 2012). As suggested by the reviewer we have clarified this point in the revised version of the manuscript (L79-83).

Comment 3 Introduction

Please clarify what are the potential “drug interactions” for an siRNA approach.

We apologize for not explaining properly that mentioned drug interactions were related to the use of small molecules for pharmacological GO inhibition and not for siRNA approaches. We have modified the sentence in the revised version of the manuscript (L104-105) to clarify this point including a new reference that supports the concerns (Cavaco and Goncalves, 2017).

Comment 4 Introduction

Although the effects of the CRISPR/Cas 9 seem long term, it's not clear one can claim it's permanent.

Genome editing systems based on sequence-specific nucleases, like the CRISPR/Cas9 system, introduce modifications in the genomic DNA that can be transferred from cell to cell. Thus, in this sense it might result in a permanent effect in the host. However, it is true that in the

absence of a selective advantage, this effect can be lost by allelic recombination or sporadic mutation. But in quiescent cells, most likely, this effect will be maintained up to cell death.

In our particular study we have seen complete GO inhibition at both early-time (Figure 1d and 1e of the manuscript) and long-time (Figure D) after therapeutic vector administration. According to previous reports, near-complete turnover of the hepatocyte mass takes place within 6 months in mouse liver-homeostasis (Furuyama et al., 2011), indicating that our CRISPR/Cas9-mediated SRT would be most likely inducing a permanent GO inhibition. Nevertheless, in the revised version of the manuscript we have replaced “permanent inhibition” by “long-term inhibition” (L130-131).

Figure D. Analysis of GO protein levels by western blot in PH1 animals treated with saline, Cas9, *Hao1-g1* or *Hao1-g2* and sacrificed 4 months after treatment.

Comment 5 Results

Please provide some semi quantitative or quantitative analysis of the renal histology.

According to the reviewer’s suggestion, we have included quantitative data of nephrocalcinosis in the revised version of the manuscript (L216-220). Three cortical areas were photographed with a 5x objective lens in a microscope (Leica DMLB) fitted with polarization filters to generate a dark field in which calcium oxalate crystals are seen as bright object (new Supplementary figure 3b). Digitized images were treated with ImageJ software to calculate the percentage area above a threshold established with a normal kidney slide photographed with the same settings. Supplementary figure 3b shows representative images from the four ranks of nephrocalcinosis found: negative (0%), mild (0-0.99%), moderate (1-5%), and severe (>5% area occupied by calcium oxalate crystals).

Comment 6 Results (last sentence)

The current study does not prove this is a safe approach in humans. Indeed, since humans (unlike mice) are not inbred, the potential for unintended off target effects is possibly much greater (and perhaps more sporadic from person to person). Given the long-term effects of this treatment, indeed, this could be a great disadvantage of this approach.

Our results indicate that the CRISPR/Cas9-mediated therapeutic approach has no *in vivo* off-target effect in the predicted higher scoring regions for both *Hao1* sgRNAs in the liver. Moreover, since the expression of SaCas9 was controlled by the liver-specific thyroxine binding globulin promoter, the genome edition should be restricted to hepatocytes. We have corroborated this point by measuring the presence of indels at the on-target site in different tissues, after administration of our therapeutic vectors (supplementary table 3). These results, together with the normal liver histology, the normal biochemical parameters in serum and the

absence of inflammatory infiltrate in the liver, suggest that the CRISPR/Cas9-mediated GO inhibition approach to treat PH1 is safe, although the safety in patients cannot be fully predicted. Thus, according to the reviewer's suggestion we have modified the text, specifying that the approach is safe in mice and its safety in humans cannot be predicted (L252-254, last sentence results).

Furthermore, as the reviewer indicates, the strategy used in this study is associated with long-term expression of the Cas9 protein, but alternative strategies associated with transient expression of the Cas9 could be eventually applied.

Comment 7 Discussion

It would seem enough to point out the differences between studies to date, rather than using terms such as "extensively criticized by a number of scientists", especially since this therapy has yet to be tried in humans.

We agree with the reviewer that it would be more constructive not using terms like "extensively criticized by a number of scientists" in the discussion. Since the study mentioned has been recently retracted (<https://www.nature.com/articles/nmeth.4293>), we have modified the discussion on the revised version of the manuscript (we have eliminated from L271 to L274 of the first version of the manuscript), and this study is now not mentioned.

Comment 7 Discussion (last sentence)

The final sentence is much too definitive since the current study does not prove that this strategy is definitive or safe for human patients with PH1.

According to the reviewer's suggestion final sentence has been modified on the revised version of the manuscript (L328-L330).

Comment 8 (overall)

Please be clear throughout that this strategy (targeting GO) is likely to only be effective for PH1 (as opposed to PH2 and 3). This is implied perhaps, but not stated.

According to the reviewer's suggestion we have specify thorough all the manuscript that our CRIPSR/Cas9-mediated GO inhibition strategy applies to PH1.

Reviewer #3 - Hyperoxaluria therapy (Remarks to the Author):

Substrate reduction has previously been shown to be an effective therapy for the treatment in a mouse model of the monogenic disease, Primary Hyperoxaluria Type 1 and more recently in patients using siRNA approaches to diminish the amount of glyoxylate produced from glycolate oxidation. In this manuscript the authors describe the utility of modifying the genomic sequence of the liver-specific enzyme, glycolate oxidase (GO), to reduce oxalate production in this mouse model to near normal. CRISPR/Cas9 technology was used coupled with an AAV vector for hepatic delivery. This research suggests that this approach could potentially be used therapeutically to treat this disease in humans. Further refinements appear to be warranted before this treatment is ready for clinical trials, however.

The effect of the treatment did not produce a change in the production of GO mRNA (Fig. 1c). The reasons for the large variability between mice in each group should be explained.

We agree with the reviewer, and as previously described the mRNA is decreased when indel frequency is high (Jarrett et al., 2017). We hypothesized that the large variability of the mRNA data, which resulted in the absence of significant differences among groups, could be associated with the fact that *Hao1* mRNA levels were analyzed in the livers of animals that were challenged with EG just before sacrifice. Thus, we have repeated the experiment in animals with no challenge and we are able to observe a significant decrease in mRNA levels in the animals from *Hao1*-g1 and *Hao1*-g2 groups in comparison to the control group (Figure 1c). In addition, now, control animals (Saline and Cas9) presented more homogenous *Hao1* mRNA and GO protein levels in absence of EG challenge. These results indicate that the administration of EG led to changes in *Hao1* expression in all the animals. We have changed the Figures 1c and 1d and introduced the correct data in the revised version of the manuscript, the text has also been changed in accordance (L154-155).

There was also a variable expression of GO protein in both saline and Cas9-treated mice in Western blots with no explanation (Fig. 1d and Suppl 1e).

As described above, the EG challenge clearly altered the expression levels of *Hao1*, leading to a high variability in GO mRNA and protein levels. The analysis of these parameters in the liver of animals that were not EG-challenged revealed more homogenous data (Figure 1c and 1d of the revised version of the manuscript).

Although oxalate excretion is significantly reduced by about half in treated Agxt mice (Fig. 3b), it appears to be twice that in WT mice. A clearer presentation of the results in a larger figure may clarify this picture which is the most relevant endpoint in their studies.

We appreciate the reviewer's comment. We have included a new graph in the revised version of the manuscript (now Figure 3b), where oxalate and glycolate basal levels are represented 4 months after treatment of the animals, in absence of any EG challenge.

The Indel frequency suggests that genomic alterations occurred in only half of the hepatocytes with both guide RNAs. Determining correlations between Indel frequencies and oxalate excretion in each mouse may show how this relationship effects the treatment.

We appreciate the reviewer's comment. We have calculated the correlation between indel percentage and oxalate levels in urine at day 7 of EG challenge. Below we enclose the results for your consideration (Figure E). The figure shows a clear relationship between editing efficiency and oxalate levels in urine. Moreover, PH1 animals presenting high editing showed similar oxalate levels to those found in WT animals (red triangles). Linear regression analysis showed a significant correlation between the two parameters ($r^2=0.575$; $p=0.0004$).

Figure E. Correlation between indel percentage and oxalate levels ($\mu\text{mol}/24\text{h}$) in all the animals. Red triangles represent the levels of oxalate in WT animals. Linear regression analysis showed a significant correlation between the two parameters.

If only half of the hepatocytes are modified this would appear to be insufficient to return oxalate production to normal.

We absolutely agree with the reviewer and we were also very surprised when we analyzed protein expression by Western blot or by immunohistochemistry showing a lack of protein expression (which explained the return of oxalate production to normal) having only 50% of indels. In this sense we have performed several experiments to address this question:

i) First, an experiment to determine the sensitivity of our western blot assay was performed, analyzing liver extracts obtained from *Hao1*^{-/-} mice mixed with liver protein extracts from WT mice at different proportions. By western blot we are able to detect GO expression even when only 10% of the cells express normal levels of the protein, indicating that our CRISPR/Cas9 treatment was able to induce a reduction of at least 90% of protein expression.

ii) Second, since Cas9 is under the control of a liver specific promoter, it is important to take into account that the CRISPR/Cas9 system will not be active in the rest of the liver cells and the editing efficiency on the hepatocytes (performed in total liver gDNA extract) might be underestimated. Thus, we have evaluated the genome editing efficiency and variant distribution by NGS in purified hepatocytes. Matched samples showed increased indel percentage when

analyzed in purified hepatocytes (new Supplementary figure 1e). We have included these results as a new Supplementary figure 1e in the revised manuscript.

Taken together, the fact that our western assay does not detect about 10% cells expressing GO and the increased hepatocyte-specific gene targeting, would explain that our CRISPR/Cas9-mediated SRT is sufficient to reduce urine oxalate levels to normal. These results have been included in the revised version of the manuscript (L167-169 and L274-276).

Cas9 treatment (Fig. 1b) suggests that it causes a low number of indels in 2 of 5 mice. Is this "noise" or to be expected? No changes were detected in Fig. 2.

We appreciate the comment, and as the reviewer suggested, the indels observed in Cas9 group using TIDE analysis are due to "noise" introduced by Sanger sequencing, in which the technique is based. In addition, the analysis of indels by NGS in the livers of animals from Cas9 groups never revealed any sign of activity in the on-target region (Figure 2, Table 1, and Supplementary figure 4).

Fig. 2 is duplicated in the Suppl figures as Fig. 4.

Figure 2 and Supplementary figure 4 represent the data obtained by NGS about indel frequency and its characteristics in the on-target region from 2 different experiments. Figure 2 corresponds to the analysis performed 1 month after the treatment of the animals, meanwhile Supplementary figure 4 shows the data analyzed 6 months after the treatment. The similarity between graphs shows the amazing reproducibility of editing in the on-target region using the same gRNAs.

The frequency of Hao1 deficiency in the population is not known, possibly because of its mild metabolic effects. Because of the paucity of individuals studied it is premature, however, to conclude the consequences are benign as stated on L87.

Asymptomatic glycolic aciduria has been reported in two brothers as the only consequence of lack of GO activity due to loss-of-function mutations at the *HAO1* gene (Frishberg et al., 2014). In addition, a large number of missense polymorphisms, many with predicted deleterious consequences at the protein level, have been found at *HAO1* in the general population (https://www.ncbi.nlm.nih.gov/SNP/snp_ref.cgi?locusId=54363). Also, a nonsense mutation (rs181543806) has been found with 0.0048 allele frequency among the Southern Han Chinese (CHS sample of the 1000 genome project). No phenotypic consequences have been detected in *Hao1*^{-/-} mice either (Martin-Higueras et al., 2016). An ongoing clinical trial targeting *HAO1* mRNA with siRNA has reported no significant safety concerns either (<https://www.businesswire.com/news/home/20180608005154/en/>). Thus, it seems likely that GO reduction in humans is safe, but we have reworded the sentence in the revised version of the manuscript (L91-92), following the reviewer's advice.

The large variations in ALT and Bilirubin in Fig. 4b indicate that in this model these assays are not reliable indicators of hepatotoxicity. H&E stains of liver sections would also be helpful to indicate the presence or absence of inflammation.

Thank you for the comment and we agree with the observation made by the reviewer. We have analyzed the liver histology of control and treated animals (hematoxylin-eosin staining) and the inflammatory infiltrate (CD45 IHC). We have modified the original Figure 4 (new Figures 4a and 4b) to include these data in the revised version of the manuscript, and the results have been commented in the text (L245-248).

The inconsistencies in the data identified above require more explanation and need to be identified as limitations.

As suggested by the reviewer we have revised the inconsistencies described above and we have included the required explanations in the revised version of the manuscript.

Cited bibliography

- Bergstralh, E.J., Monico, C.G., Lieske, J.C., Herges, R.M., Langman, C.B., Hoppe, B., Milliner, D.S., and Investigators, I. (2010). Transplantation outcomes in primary hyperoxaluria. *Am J Transpl.* 10, 2493–2501.
- Cameron, P., Fuller, C.K., Donohoue, P.D., Jones, B.N., Thompson, M.S., Carter, M.M., Gradia, S., Vidal, B., Garner, E., Slorach, E.M., et al. (2017). Mapping the genomic landscape of CRISPR-Cas9 cleavage. *Nat. Methods* 14, 600–606.
- Cavaco, M., and Goncalves, J. (2017). Interactions Between Therapeutic Proteins and Small Molecules: The Shared Role of Perpetrators and Victims. *Clin. Pharmacol. Ther.* 102, 649–661.
- Cochat, P., and Rumsby, G. (2013). Primary Hyperoxaluria. *N. Engl. J. Med.* 369, 649–658.
- Cochat, P., Hulton, S.-A., Acquaviva, C., Danpure, C.J., Daudon, M., De Marchi, M., Fargue, S., Grothoff, J., Harambat, J., Hoppe, B., et al. (2012). Primary hyperoxaluria Type 1: indications for screening and guidance for diagnosis and treatment. *Nephrol. Dial. Transplant.* 27, 1729–1736.
- Edvardsson, V.O., Goldfarb, D.S., Lieske, J.C., Beara-Lasic, L., Anglani, F., Milliner, D.S., and Palsson, R. (2013). Hereditary causes of kidney stones and chronic kidney disease. *Pediatr. Nephrol.* 28, 1923–1942.
- Fargue, S., Harambat, J., Gagnadoux, M.-F., Tsimaratos, M., Janssen, F., Llanas, B., Berthélémy, J.-P., Boudailliez, B., Champion, G., Guyot, C., et al. (2009). Effect of conservative treatment on the renal outcome of children with primary hyperoxaluria type 1. *Kidney Int.* 76, 767–773.
- Frishberg, Y., Zeharia, A., Lyakhovetsky, R., Bargal, R., and Belostotsky, R. (2014). Mutations in HAO1 encoding glycolate oxidase cause isolated glycolic aciduria. *J. Med. Genet.* 51, 526–529.
- Furuyama, K., Kawaguchi, Y., Akiyama, H., Horiguchi, M., Kodama, S., Kuhara, T., Hosokawa, S., Elbahrawy, A., Soeda, T., Koizumi, M., et al. (2011). Continuous cell supply from a Sox9-expressing progenitor zone in adult liver, exocrine pancreas and intestine. *Nat. Genet.* 43, 34–41.
- Harambat, J., van Stralen, K.J., Espinosa, L., Grothoff, J.W., Hulton, S.-A., Cerkauskiene, R., Schaefer, F., Verrina, E., Jager, K.J., Cochat, P., et al. (2012). Characteristics and outcomes of children with primary oxalosis requiring renal replacement therapy. *Clin. J. Am. Soc. Nephrol.* 7, 458–465.
- Hopp, K., Cogal, A.G., Bergstralh, E.J., Seide, B.M., Olson, J.B., Meek, A.M., Lieske, J.C., Milliner, D.S., and Harris, P.C. (2015). Phenotype-Genotype Correlations and Estimated Carrier Frequencies of Primary Hyperoxaluria. *J. Am. Soc. Nephrol.* 26, 2559–2570.
- Hoyer-Kuhn, H., Kohbrok, S., Volland, R., Franklin, J., Hero, B., Beck, B.B., and Hoppe, B. (2014). Vitamin B6 in primary hyperoxaluria I: First prospective trial after 40 years of practice. *Clin. J. Am. Soc. Nephrol.* 9, 468–477.
- Jarrett, K.E., Lee, C.S.C.M., Yeh, Y.-H.H., Hsu, R.H., Gupta, R., Zhang, M., Rodriguez, P.J., Lee, C.S.C.M., Gillard, B.K., Bissig, K.-D.D., et al. (2017). Somatic genome editing with CRISPR/Cas9 generates and corrects a metabolic disease. *Sci. Rep.* 7, 44624.
- Leumann, E., and Hoppe, B. (2001). The Primary Hyperoxalurias. *J. Am. Soc. Nephrol.* 12, 1986–1993.
- Martin-Higueras, C., Luis-Lima, S., and Salido, E. (2016). Glycolate oxidase is a safe and efficient target for substrate reduction therapy in a mouse model of primary hyperoxaluria Type I. *Mol. Ther.* 24, 719–725.
- Tsai, S.Q., Nguyen, N.T., Malagon-Lopez, J., Topkar, V. V., Aryee, M.J., and Joung, J.K. (2017). CIRCLE-seq: A highly sensitive in vitro screen for genome-wide CRISPR-Cas9 nuclease off-targets. *Nat. Methods* 14, 607–614.
- van Woerden, C.S., Grothoff, J.W., Wanders, R.J.A., Davin, J.C., and Wijburg, F.A. (2003). Primary hyperoxaluria type 1 in The Netherlands: Prevalence and outcome. *Nephrol. Dial. Transplant.* 18, 273–279.

Reviewer #1 (Remarks to the Author):

The authors addressed all my comments and the data is robust and convincing now. While the data is convincing the discussion lacks a bit the broader picture and is focused only on PH1. Giving a little bit more background (instead of only listing references) and also mentioning 1-2 potential applications of the therapeutic concept would increase the manuscript and help the readers.

Reviewer #2 (Remarks to the Author):

My concerns have been adequately addressed. Thanks!

Reviewer #3 (Remarks to the Author):

The authors have adequately answered all of the concerns raised by the reviewers and I believe it is now suitable for publication.

Point by point response to the reviewers

(manuscript NCOMMS-18-02083A by Zabaleta *et al.*)

We thank the positive comments of three reviewers. Again, we would like to express our gratitude to the reviewers for their excellent suggestions on the previous version of the manuscript. Below you can find the detailed answer to the issue raised by the reviewers.

Reviewer #1 (Remarks to the Author):

The authors addressed all my comments and the data is robust and convincing now. While the data is convincing the discussion lacks a bit the broader picture and is focused only on PH1. Giving a little bit more background (instead of only listing references) and also mentioning 1-2 potential applications of the therapeutic concept would increase the manuscript and help the readers.

Response: We are very grateful to the reviewer, since the quality of the manuscript substantially improved following the reviewer's advice. Following his/her suggestions we have modified the introduction (L110-L112) and the discussion sections (L254-L259 and L331-L342) to include more detailed background and potential applications of the therapeutic approach described in this work. We believe that this revised version of the manuscript has been improved following the reviewers' advice, including a broader view of the therapeutic concept of our study.

Reviewer #2 (Remarks to the Author):

My concerns have been adequately addressed. Thanks!

Response: As with the first reviewer, we are very grateful because, thanks to his/her advice, a stronger version of the manuscript has been submitted.

Reviewer #3 (Remarks to the Author):

The authors have adequately answered all of the concerns raised by the reviewers and I believe it is now suitable for publication.

Response: We are also very grateful with the third reviewer since reviewer's suggestions and comments substantially improved the manuscript.